

# Neighbourhood socioeconomic characteristics and blood pressure among Jamaican youth: a pooled analysis of data from observational studies

Trevor S. Ferguson[1], Novie O.M. Younger-Coleman[1], Jasneth Mullings[2], Damian Francis[3], Lisa-Gaye Greene[4], Parris Lyew-Ayee[4] and Rainford Wilks[1]

[1] Epidemiology Research Unit, Caribbean Institute for Health Research, The University of the West Indies, Mona, Kingston, Jamaica
[2] Health Research Resource Unit, Dean's Office, Faculty of Medical Sciences, University of the West Indies, Mona, Kingston, Jamaica
[3] School of Health and Human Performance, Georgia College and State University, Milledgeville, GA, United States of America
[4] Mona GeoInformatics Institute, The University of the West Indies, Mona, Kingston, Jamaica

Corresponding author
Trevor S. Ferguson,
trevor.ferguson02@uwimona.edu.jm

## ABSTRACT

**Introduction**. Neighbourhood characteristics are associated with several diseases, but few studies have investigated the association between neighbourhood and health in Jamaica. We evaluated the relationship between neighbourhood socioeconomic status (SES) and blood pressure (BP) among youth, 15–24 years old, in Jamaica.

**Methods**. A pooled analysis was conducted using data from three studies (two national surveys and a birth cohort), conducted between 2005–2008, with individual level BP, anthropometric and demographic data, and household SES. Data on neighbourhood SES were obtained from the Mona Geo-Informatics Institute. Neighbourhood was defined using community boundaries from the Social Development Commission in Jamaica. Community characteristics (poverty, unemployment, dependency ratio, population density, house size, and proportion with tertiary education) were combined into SES scores using principal component analysis (PCA). Multivariable analyses were computed using mixed effects multilevel models.

**Results**. Analyses included 2,556 participants (1,446 females; 1,110 males; mean age 17.9 years) from 306 communities. PCA yielded two neighbourhood SES variables; the first, PCA-SES1, loaded highly positive for tertiary education and larger house size (higher value = higher SES); while the second, PCA-SES2, loaded highly positive for unemployment and population density (higher value = lower SES). Among males, PCA-SES1 was inversely associated with systolic BP ($\beta$-1.48 [95%CI $-2.11$, $-0.84$] mmHg, $p < 0.001$, for each standard deviation unit increase in PCA-SES1 score) in multivariable model accounting for age, household SES, study, BMI, fasting glucose, physical activity and diet. PCA-SES1 was not significantly associated with systolic BP among females ($\beta$ $-0.48$ [$-1.62$, $0.66$], $p = 0.410$) in a similar model. Associations for PCA-SES2 was assessed using linear splines to account for non-linear effects. The were no significant associations between systolic BP and PCA-SES2 among males. Among females, higher PCA-SES2 (i.e. lower SES) was associated with higher systolic BP at spline 2 [$z$-score -1 to 0] ($\beta$4.09 [1.49, 6.69], $p = 0.002$), but with lower systolic BP at spline 3 [$z$-core 0 to 1] ($\beta$-2.81 [$-5.04$, $-0.59$], $p = 0.013$). There were no significant

associations between diastolic BP and PCA-SES1, but PCA-SES2 showed non-linear associations with diastolic BP particularly among males.

**Conclusion**. Higher neighbourhood SES was inversely associated with systolic BP among male Jamaican youth; there were non-linear associations between neighbourhood SES and systolic BP among females and for diastolic BP for both males and females.

## INTRODUCTION

Neighbourhood characteristics, in particular conditions of the physical and social environment, have been shown to be associated with a number of health-related conditions (*Arcaya et al., 2016*; *Diez Roux, 2016*; *Diez Roux & Mair, 2010*; *Duncan & Kawachi, 2018*; *Oakes et al., 2015*; *Pemberton & Humphris, 2016*; *Ribeiro, 2018*). Research in this area has grown exponentially in the last twenty years, with obesity and depression being among the conditions most often studied (*Arcaya et al., 2016*; *Diez Roux, 2016*; *Diez Roux & Mair, 2010*; *Oakes et al., 2015*). Other outcomes studied include cardiovascular health, allostatic load, diabetes, dietary practices, physical activity, alcohol consumption and tobacco use (*Airaksinen et al., 2015*; *Assari et al., 2016*; *Diez Roux, 2016*; *Ribeiro et al., 2019*; *Unger et al., 2014*). Neighbourhood characteristics assessed include poverty, deprivation, walkability, food environment, healthy food availability, fast food density, air pollution, traffic noises, social cohesion, crime, victimization, fear of neighbourhood violence, and levels of urbanization, among others (*Airaksinen et al., 2015*; *Arcaya et al., 2016*; *Assari et al., 2016*; *Diez Roux & Mair, 2010*; *Jivraj et al., 2019*; *Ribeiro et al., 2019*; *Unger et al., 2014*). In general, these research studies have found that poorer neighbourhood conditions are associated with worse health outcomes (*Diez Roux, 2016*).

Underlying mechanisms possibly include interactions between adverse environmental exposures, reduced sense of safety due to potential exposure to crime and violence, increase levels of adverse health behaviours and increased psychosocial stress (*Diez Roux & Mair, 2010*). The effects of neighbourhood on health can be conceptualized as being either compositional or contextual (*Macintyre & Ellaway, 2003*). The compositional effects refer to the differences in the characteristics of the people living in the neighbourhood, while the contextual effects are due to differences between the places (*Macintyre & Ellaway, 2003*). Contextual effects include collective social functioning and social practices (*Macintyre, Ellaway & Cummins, 2002*). Overall, McIntyre and colleagues have proposed that features of the local area that influence health include: physical features of the environment, availability of healthy environments, services provided to support people in their daily lives, sociocultural features of the neighbourhood and the reputation of the area (*Macintyre, Ellaway & Cummins, 2002*).

With regards to blood pressure, associations have been reported for systolic blood pressure, diastolic blood pressure and hypertension (*Coulon et al., 2016*; *Fan et al., 2015*; *Liu et al., 2013*), using indicators such as community level income, poverty, social and environmental characteristics and 'perceived neighbourhood crime and satisfaction'. Additionally, associations have been reported between attenuation of the normal nocturnal decrease in blood pressure (dipping) and higher levels of perceived neighbourhood problems (*Euteneuer et al., 2014*). Studies have also shown associations between neighbourhood characteristic and blood pressure in youth (*McGrath, Matthews & Brady, 2006*). Possible mechanisms contributing to the effects of neighbourhood on blood pressure include poor dietary behaviour, increased obesity, higher consumption of alcohol and increased psychosocial stress (*Chaix et al., 2010*; *Chaix et al., 2008*; *Euteneuer et al., 2014*; *Mujahid et al., 2011*).

In Jamaica, research on the association between neighbourhood and health is limited. Cunningham-Myrie and colleagues found associations between characteristics of the neighbourhood environment and physical activity and obesity, with significant differences by sex (*Cunningham-Myrie et al., 2015*), while Mullings and colleagues found neighbourhood associations with depressive symptoms, again with significant sex differences (*Mullings et al., 2013*). Two other studies were identified, one reporting an association between neighbourhood crime and childhood malnutrition (*Thompson et al., 2017*) and the other reporting associations between perceived neighbourhood characteristics and depressive symptoms among adolescents in Jamaica and other Caribbean islands (*Lowe et al., 2014*). More recently, associations have been reported between neighbourhood disorder and cumulative biological risk and substance use (*Cunningham-Myrie et al., 2018*; *Felker-Kantor et al., 2019*).

To our knowledge, no previous study on the relationship between blood pressure and neighbourhood characteristics has been conducted in Jamaica. Research on the neighbourhood effects on blood pressure is also limited in developing countries, among black populations and among youth. Studies have shown that social determinants of health often vary between countries and in different social context (*Leng et al., 2015*; *Mackenbach et al., 2008*), suggesting that findings are often not generalizable across countries or regions. Given the high burden of hypertension in developing countries and black populations (*Gakidou et al., 2017*; *NCD Risk Factor Collaboration, 2017*), and the fact that blood pressure tracks across the life course from childhood into adulthood (*Chen & Wang, 2008*), studies on the neighbourhood effects of blood pressure in these settings and among youth will further enhance our understanding and inform public health decisions. In light of the foregoing, this study forms part of a larger body of work, exploring social and biological determinants of blood pressure in adolescents and young adults in a predominantly black population from a developing country (*Ferguson et al., 2015*; *Ferguson et al., 2018*). Previous studies evaluated individual and household socioeconomic status but did not address the neighbourhood context.

This study therefore aimed to:
1. Evaluate the relationship between neighbourhood characteristics and systolic and diastolic blood pressure among Jamaican youth, 15–24 years old.

2. Evaluate whether the presence of elevated blood pressure or hypertension (BP $\geq$ 120/80 mmHg) is associated with neighbourhood characteristics.

Estimates will adjust for potential confounders including age, sex and household socioeconomic status and will also evaluate the effect of possible intermediary variables and independent covariates. We hypothesized that lower neighbourhood SES will be associated with higher systolic and diastolic blood pressure and higher prevalence of elevated blood pressure or hypertension. We chose to assess elevated blood pressure or hypertension as our high-risk category since the study involved adolescents and young adults and prevalence of hypertension would be relatively low in this age-group.

## METHODS

### Data sources

The study was conducted using data from three studies (two cross-sectional surveys and one birth cohort) which included measurements of blood pressure for youth aged 15–24 years. We obtained data on neighbourhood characteristics from the Mona Geo-Informatics Institute Geographic Information Systems (GIS) Database. The studies included are (1) the 1986 Jamaica Birth Cohort Study 18–20 Follow Up (1986-JBCS), (2) The Jamaica Youth Risk and Resiliency Behaviour Survey 2006 (JYRRBS) and (3) The Jamaica Health and Lifestyle Survey 2007–2008 (JHLS-II). All studies were conducted by the Epidemiology Research Unit with one of the investigators (RJW) as the principal investigator for each study. All three studies were previously approved by the University of the West Indies Ethics Committee (ethics approval numbers: 1986 JBCS—no number, letter dated June 1, 2005, JYRRBS—ECP 71, 2005/2006, JHLS-II—ECP 161, 2006/2007). Additionally, the proposal for this pooled analysis was reviewed and approved by The University of the West Indies Ethics Committee (study number: ECP 173, 16/17), prior to obtaining data on neighbourhood characteristics from the Mona Geo-Informatics Institute. Participants provided written informed consent at the time of data collection for the primary studies.

Details on the methods used in these studies have been previously published (*Ferguson et al., 2010a*; *McCaw-Binns et al., 2011*; *Wilks et al., 2007*; *Wilks et al., 2008*). We therefore present only a brief description of each study and the Mona Geo-Informatics Institute.

The 1986-JBCS third follow up (*Ferguson et al., 2010a*; *McCaw-Binns et al., 2011*) was conducted between 2005 and 2007 and included 902 young adults who were 18-20 years old at the time of evaluation. Response rate was 74% among contacted participants and 55% of targeted participants (*Ferguson et al., 2010a*). Participants were part of a longitudinal study of children born in Jamaica during the months of September and October of 1986 and were originally part of the Jamaica Perinatal Mortality Survey. Measurements included blood pressure, anthropometry, individual and household socioeconomic status as well a variety of laboratory markers of chronic disease.

JYRRBS (*Wilks et al., 2007*) was a national cross-sectional study, conducted in 2006, and included 1317 participants 15–19 years old. Response rate for this study was 99% (*McFarlane et al., 2014*). The study evaluated lifestyle and behaviour risk factors for chronic non-communicable disease, including mental health, reproductive and sexual health behaviours and factors which conferred resilience against adverse health behaviours.

JHLS-II (*Wilks et al., 2008*) was also a national health examination survey, conducted between 2007 and 2008 and included 2848 participants 15–74 years old. Response rate for this study was approximately 98% (*Ferguson et al., 2017*). The study collected data on chronic disease risk factors, health behaviours and socioeconomic status. JHLS-II included 520 persons who were 15–24 years old, who were therefore eligible for inclusion in this study.

The Mona Geo-Informatics Institute (MGI) is an Institute of The University of the West Indies, Mona, and provides several GIS resources to the University and other stakeholders. MGI has an extensive human and social mapping database which includes geo-spatial data on environmental characteristics, infrastructure and social characteristics, at the parish, constituency, electoral division or community level. Data are derived from population censuses, government databases, non-governmental organizations, private sector partners and international organizations. These data are linked geo-spatially to MGI's extensive mapping of the island. Additional information is available on the MGI website (http://www2.monagis.com/). For this study, MGI provided community data for anonymized participants from the three studies. We provided MGI with each participant's address or electoral district, which were used to map them in communities as defined by The Planning Institute of Jamaica and the Social Development Commission (*Planning Institute of Jamaica, 2006*). Data on community characteristics were then merged with data from each study. Each participant therefore had associated community level data which were used for analyses.

## Measurements

The study used previously collected data from the studies described above and from MGI databases. Data collection for all the included studies followed similar standardized protocols. In all studies, blood pressure was measured using a mercury sphygmomanometer (W.A. Baum & Co; New York) after the participant had been seated for five minutes. The mean of the second and third systolic and diastolic blood pressures were used for the analysis. Elevated blood pressure or hypertension was defined as systolic blood pressure $\geq$120 mmHg or diastolic blood pressure of $\geq$80 mmHg or being on medication for hypertension. This categorization includes persons classified as prehypertension or hypertension in The Seventh Report of the Joint National Committee on Prevention, Detection, Evaluation, and Treatment of High Blood Pressure and as elevated blood pressure or hypertension in the 2017 American College of Cardiology / American Heart Association Guidelines (*Chobanian et al., 2003*; *Whelton et al., 2018*). This cut-point was used given that the prevalence of hypertension would be relatively low in the age-group included in the study, and the fact that studies have shown that blood pressure levels in the range 120–139 mmHg systolic and 80–89 mmHg diastolic are associated with higher cardiovascular risk (*Ferguson et al., 2008*; *Lewington et al., 2002*; *Vasan et al., 2001*). Weight was measured using a portable digital scale (Tanita brand, Tokyo, Japan) and height measured with a portable stadiometer (Seca, Hamburg, Germany). Body mass index (BMI) was calculated by dividing the weight (in kilograms) by the square of height (in meters). Study specific internal BMI $z$-scores were obtained and used for bivariate

and multivariable analyses. Waist and hip circumferences were measured with a non-stretchable nylon tape measure. Data on individual and household socioeconomic status were collected using interviewer administered questionnaires. Household socioeconomic status was assessed using number of reported household assets, categorized into thirds. Data on physical activity were obtained using a locally developed questionnaires for the 1986-JBCS and JHLS-II and the International Physical Activity Questionnaire short form (https://sites.google.com/site/theipaq/) for the JYRRBS. For each study, participants were classified as having high, moderate or low physical activity levels. Classification rules for the locally developed questionnaires have been previously published (*Ferguson et al., 2011*; *Ferguson et al., 2010a*), while the classifications used in JYRRBS can be found at the IPAQ support site (https://sites.google.com/site/theipaq/scoring-protocol). We did not have data on salt intake and therefore frequency of fast food consumption was used as a surrogate for high salt diet. Data on fast food consumption was collected by questionnaire and categorized as less than twice per week, 2 to 4 times per week or five or more times per week.

As stated above, data for neighbourhood characteristics were collated by MGI from population censuses, various governmental and non-governmental organizations, and private sector partners. We opted to use these data to evaluate neighbourhood socioeconomic status because there were no generally accepted measures for use in Jamaica. Previous studies have used indexes based on aspects of the built environment, services, poverty, education and crime (*Cunningham-Myrie et al., 2015*; *Mullings et al., 2013*), however some of these variables were only available for JHLS-II. Details on the variables used in this paper and data sources are provided in Table S1. Derivation of the community socioeconomic status scores are described below under statistical methods.

For these analyses, neighbourhood was defined as the community in which the participants lived. In Jamaica, community boundaries are defined by the Social Development Commission (website: https://sdc.gov.jm/) as geographic areas grouping people based on common ownership of resources or sharing of social, economic or cultural facilities. The country is thus divided into parishes, developmental areas, communities and districts. Jamaica has 14 parishes. In its most recent listing, the SDC social boundaries include 775 communities. For the communities included in this study, population ranged from just over 100 to over 60,000 community members. Further description of the community characteristics is shown in Table S2.

## Sample size and power

Given that the analyses were conducted using previously collected data, we performed sample size calculations to assess if the available data were adequate to answer the primary research objective, i.e., association between systolic blood pressure and community SES score. Calculations were based on the observed correlation between systolic blood pressure and the neighbourhood socioeconomic status score (described below) and were performed for males and females separately. Correlation coefficients were −0.15 for females and −0.14 for males. Using Stata's '*power onecorrelation*' command, with $\alpha = 0.05$ and power of 80%, we estimated that the required sample size for a simple random sample for females was

340 and for males 421. In order to account for clustering of observations by community as used in the analyses (see below) we applied a design effect using the formula *design effect* $= 1 + (M - 1) \times ICC$, where M = number of observations per cluster and ICC = intraclass correlation coefficient (*Lohr, 1999*). The average number of observations per community was obtained using Stata's *xtsum* command; the ICCs were obtained from sex-specific mixed effects multi-level models with systolic blood pressure as the outcome variable and neighbourhood socioeconomic status score as the explanatory variable; cluster variables were parish and community. For females, the $M = 6$ and ICC = 0.134. For males, $M = 5$ and ICC = 0.087. These calculations yielded a design effect of 1.67 for females and 1.35 for males, so that the required sample size for females of 569 and for males 568. The available sample in our pooled analyses included 1,446 females and 1,110 males and would have adequate power for our primary analyses. Details of the sample size calculations are available in Supplemental Information 2.

## Statistical methods

Analyses were performed using Stata 14.2 (StataCorp, College Station, Texas). Individual datasets were prepared for each study, and variables recoded to ensure that codes were the same for variables to be used for this analysis. The three datasets were then merged into a single dataset with a variable to indicate the study from which each participant came. Descriptive statistics were then obtained, initially stratified by study, and then stratified by study and sex. Summary statistics were also obtained for all participants stratified by blood pressure category. We also performed analyses to assess whether there was evidence for additive interaction by sex and by study in the relationship between systolic BP and neighbourhood SES and other covariates included in the models. For all analyses, a *p*-value <0.05 was considered statistically significant.

   Principal Component Analysis (PCA) was used to derive composite socioeconomic status (SES) variables. We initially assessed the correlation between the available SES variables and used variables with significant correlations for consideration in the PCA. We then used the Kaiser-Meyer-Olkin (KMO) test for sampling adequacy and the Bartlett test of sphericity to select the variables for inclusion in final PCA model (*Hair et al., 2006*). We aimed to choose a combination of variables with KMO >0.5 and Bartlett test for sphericity with $p < 0.05$ (thus rejecting the null hypothesis that variables are uncorrelated). Based on these criteria, a final list of six community variables were chosen for inclusion in the PCA model (KMO = 0.774; Bartlett's test for sphericity $p < 0.001$). These variables were: percent poverty, percent unemployment, dependency ratio, population density, percent of houses with two or more bedrooms, and proportion with tertiary education. We believe that these six variables had good content validity, representing aspects of income, education, employment, crowding and wealth (size of house) which are all factors related to neighbourhood socioeconomic position. Horn's parallel analysis method was used to select components to be used in the analysis; components having an adjusted eigenvalue greater than 1.0 used in the analyses (*Dinno, 2009*; *Hayton, Allen & Scarpello, 2004*). This method yielded two PCA components which were then used to predict SES scores for each community. A graph showing the observed and adjusted eigenvalues is shown in

Fig. S1. Factor loadings for each variable included in the PCA components are shown in Table S3. We assessed face validity by checking the component scores against perceived social status of the communities based on the investigators' understanding of social class in Jamaica. Analyses were conducted using these PCA derived community SES scores treated as standardized $z$-scores or categorized into SES thirds. We also evaluated whether the relationship between systolic BP and the PCA-SES variables showed evidence for non-linear effects, by adding a quadratic term to the model. Where there was evidence for a non-linear effect, we used linear splines in the bivariate and multivariable models. Linear splines are used to estimate the relationship between outcome and exposure as a piecemeal linear function, with the exposure variable divided into linear segments joined at selected points called knots (*Harrell, 2015*; *StataCorp, 2015a*).

We used multilevel regression models to evaluate the association between community SES and blood pressure. Analyses with systolic and diastolic blood pressure as outcomes were done using mixed effect linear regression models with maximum likelihood (ML) estimation, while analyses with elevated blood pressure as the outcome were done with mixed effects logistic regression models. The multi-level model accounted for individuals nested within communities, and communities nested within parishes. The overall multilevel model structure was therefore: level 1—individual study participants; level 2—communities in which participant lived; level 3—parish in which community was situated. We report fixed effects coefficients adjusted for clustering by parish and community.

Multiple imputation by chained equations was used to account for missing data. Analyses were restricted to participants who had available data on the outcome variables, systolic and diastolic blood pressure, but missing data for all other variables were imputed when necessary. Eight variables had between 1 and 144 missing values, with the majority of variables having 10 or fewer missing values. Overall, 200 participants (7.8% of total) had at least one missing value. Table S4 presents a listing of the number and percentage of missing values for variables with missing values included in the analyses. A comparison of mean values for characteristics among complete cases versus those with missing values revealed no statistically significant differences; however, there was a statistically significant difference in the proportion of persons within thirds of the second PCA SES variable ($\chi^2 = 8.68$, $p = 0.013$). Based on this we assessed the data to be missing at random (MAR) (*White, Royston & Wood, 2011*). Using multiple imputation would therefore reduce potential bias associated with the complete case analysis and improve the power of the study (*Nguyen, Carlin & Lee, 2017*; *White, Royston & Wood, 2011*). A stacked multiple imputed dataset comprising the original data set and 20 datasets with imputed values was therefore created using Stata's mi suite of commands (*StataCorp, 2015b*). Given that we had less than 10% of data missing, we chose to do 20 imputations as recommended by Graham and colleagues (*Graham, Olchowski & Gilreath, 2007*). Multiple imputation by chained equations was performed using Stata's *mi impute chained* command (*StataCorp, 2015b*). The imputation model included the outcome variables (systolic and diastolic blood pressure), the exposure variables (PCA-SES standardized scores), sex, and all variables for which imputation were performed (age, BMI, height, fasting glucose, fasting cholesterol, household possession, fast food consumption, and physical activity levels). The syntax for the imputation models

is shown in Supplemental Information 2. The imputed dataset was used for bivariate and multivariable models using Stata's *mi estimate* commands; these commands run estimates on each of the imputed datasets and combine estimates using Rubin's rules (*Marshall et al., 2009*; *StataCorp, 2015b*). For model building we extracted the first imputed data set and performed regular (non-imputed) regression analyses (*Wood, White & Royston, 2008*).

For multivariable analyses, we developed sequential models based on our hypothesized associations, in order to assess the effect of potential confounders, intermediary variables or independent covariates. Age, sex and household socioeconomic status were treated as potential confounders, while BMI, glucose, physical activity and diet were considered possible intermediary variables or independent covariates. A directed acyclic graph illustrating the hypothesized association is shown in Fig. S2. Sex-specific models were used as there was evidence for interaction by sex for some variables (see details in results section and in Table S5). Model 1 included the outcome variable and exposure variables along with age and household SES as potential confounders. We also included a variable for study in this model to account for differences due to the specific study participants were from. For model 2, BMI was added to the model, and for model 3 we added glucose, physical activity and fast food consumption. We opted to keep all the hypothesized variables in the full model so as to avoid potential biases from variable selection (*Greenland & Pearce, 2015*). Based on the number of participants and the number of variables in the full model, we had more than the 10–15 observations per covariate coefficient for linear models as recommended by *Babyak (2004)*. For binary models, we had more than four subjects per confounder coefficient as recommended by Greenland, and more than ten events per variable as recommended by Peduzzi (*Greenland, Daniel & Pearce, 2016*; *Greenland & Pearce, 2015*; *Peduzzi et al., 1996*). Additionally, there was no evidence for multicollinearity for the variables included. Final models were run using the 20 stacked multiple imputation datasets for reporting. We also reran the models using complete cases only to assess whether our results were influenced by the imputation. Model assessment was done by checking whether there was evidence for collinearity or heteroscedasticity and whether the distribution of standardized residuals deviated from the normal distribution assumption. We also assessed whether multivariate outliers had any meaningful impact on the models. Where there was evidence of heteroscedasticity, we used robust standard errors (*Hamilton, 2013*).

## RESULTS

Final analyses included 2,556 participants (1,446 females; 1,110 males) from 306 communities. Mean age was 17.9 years (standard deviation 2.0). There was evidence for significant clustering of both systolic BP and diastolic BP within communities, with intra-class correlation coefficients of 10.7% (95% CI [6.8–16.6%]) and 10.9% (95% CI [7.0–16.6%]), respectively.

PCA yielded two components with adjusted eigenvalues >1.0 and these were used to derive two neighbourhood SES variables. These two variables were standardized using $z$-scores for analyses. The first component, PCA-SES1, had an adjusted eigenvalue of 3.21

PeerJ _____________________________________

and explained 54% of the variance; this component loaded highly positive for tertiary education and larger house size (higher value = higher SES). The second component, PCA-SES2, had an adjusted eigenvalue of 1.40 and explained 24% of the variance; this component loaded highly positive for unemployment and population density (higher value = lower SES). Figure S1 shows the plot for the observed and adjusted eigenvalues for the principal components, while Table S3 shows the eigenvectors (factor loadings) for the individual variables included in PCA-SES1 and PCA-SES2. The investigators assessed the PCA-SES components as having good face validity, in that communities with high scores for PCA-SES1 were those generally accepted as high SES in the local context and those with high scores on PCA-SES2 were those accepted as having lower SES.

Table 1 shows summary statistics for characteristics of all study participants combined and for each of the three studies included in the analysis. Mean age ranged from 16.5 to 19.8 years and was highest for JHLS-II. While there were statistically significant differences for participant characteristics between studies, most of these differences were small, except for waist circumference which showed an approximately six-point difference, with mean value of 77.1 cm in JHLS-II compared to 71.6 in JYRRBS. Mean $z$-score for PCA-SES 1 was highest in the 1986-JBCS (0.37) and lowest in the JYRRBS (-0.12), suggesting that participants from the 1986-JBCS had higher overall SES. This was supported by the distribution of individual community SES characteristics as shown in Table S2. Participants in the 1986-JBCS came from communities with larger populations, greater population density, less poverty, and lower dependency ratio. Mean $z$-score for PCA-SES 2 was also highest in the 1986-JBCS. Prevalence of elevated blood pressure or hypertension was 29% overall and ranged from 21% in the 1986-JBC to 37% in JHLS-II (see Table 1). There were also significant differences in the distribution of persons in BMI categories, thirds of household possessions, fast food consumption categories and physical activity levels. Additional data with summary statistics for participant characteristics, stratified by study and sex, are shown in Table S6. There were significant sex differences for most of the participant characteristics.

Table 2 shows the means or percentages for participant characteristics within blood pressure categories, defined as normal (<120/80 mmHg) and elevated BP or hypertension (≥120/80 mmHg). Participants with elevated BP or hypertension were taller and heavier and had higher mean BMI and waist circumference. For the categorical variables, elevated BP was significantly associated with sex, BMI category, and physical activity level.

As stated before, we assessed whether the relationship between systolic BP and the PCA-SES variables showed evidence for non-linear effects by adding a quadratic term to the model. This was non-significant for PCA-SES1, but significant for PCA-SES2 ($p < 0.001$); we therefore used linear splines in the bivariate and multivariable analyses for PCA-SES2. We also assessed for interaction by sex and study. Tests for sex interaction were significant for the relationship between systolic BP and some covariates (age, BMI, PCA-SES2 at spline 4, and fast food consumption category 3–4 times/week), therefore we present sex-specific analyses for bivariate and multivariable models. Details of the sex interaction effects are shown in Table S5. There was no evidence for interaction by study in the relationship between systolic BP and PCA-SES1. For PCA-SES2, there was some evidence for interaction at spline 1 and 4 for females and spline 3 for males (see Table S7).

**Table 1** Summary statistics for participant and community characteristics by study.

| Characteristics | All Participants N=2556 | 1986 Birth Cohort n = 893 | Youth Risk Survey n = 1200 | Jamaica Health & Lifestyle Survey n = 463 | P-value |
|---|---|---|---|---|---|
| | Mean (SD) | Mean (SD) | Mean (SD) | Mean (SD) | |
| Age (years) | 17.9 (2.0) | 18.8 (0.6) | 16.5 (1.3) | 19.9 (2.7) | <0.001 |
| Height (cm) | 169.0 (9.4) | 169.6 (9.1) | 167.0 (9.7) | 167.4 (8.8) | <0.001 |
| Weight (kg) | 64.5 (14.6) | 66.4 (15.6) | 62.1 (13.0) | 67.1 (15.7) | <0.001 |
| Body mass index (kg/m$^2$) | 22.9 (6.1) | 23.0 (5.0) | 22.3 (4.7) | 24.0 (5.8) | <0.001 |
| Systolic BP (mmHg) | 111.5 (10.8) | 110.4 (10.1) | 111.4 (10.5) | 113.9 (12.3) | <0.001 |
| Diastolic BP (mmHg) | 69.8 (10.2) | 67.9 (9.8) | 70.4 (10.1) | 71.8 (10.5) | <0.001 |
| Waist circumference (cm) | 73.6 (10.9) | 74.5 (11.5) | 71.6 (9.0) | 77.1 (13.0) | <0.001 |
| Fasting glucose (mmol/l) | 4.0 (1.2) | 4.6 (0.5) | 3.5 (1.2) | 4.1 (1.6) | <0.001 |
| PCA-SES Component 1 (z-score) | 0.01 (1.0) | 0.37 (0.98) | −0.12 (0.99) | 0.19 (0.86) | <0.001 |
| PCA SES Component 2 (z-score) | 0.004 (1.01) | 0.08 (1.15) | −0.01 (0.93) | −0.11 (0.90) | <0.001 |
| | n (%) | n (%) | n (%) | n (%) | |
| Sex | | | | | <0.001 |
| Female | 1446 (56.6) | 485 (54.3) | 657 (54.8) | 304 (65.7) | |
| Male | 1110 (43.4) | 408 (45.7) | 543 (45.3) | 159 (34.3) | |
| Elevated BP/HTN (BP ≥ 120/80 mmHg) | 749 (29.3) | 188 (21.1) | 388 (32.3) | 173 (37.4) | <0.001 |
| Body Mass Index Category | | | | | <0.001 |
| Underweight | 322 (12.6) | 94 (10.6) | 181 (15.1) | 47 (10.3) | |
| Normal weight | 1615 (63.4) | 575 (64.5) | 775 (64.8) | 264 (57.8) | |
| Overweight | 400 (15.7) | 149 (16.7) | 169 (14.1) | 82 (17.9) | |
| Obesity | 209 (8.2) | 73 (8.2) | 72 (6.0) | 64 (14.0) | |
| Household Possessions Category | | | | | 0.012 |
| Lower Third | 913 (35.7) | 350 (39.2) | 417 (34.8) | 146 (31.5) | |
| Middle Third | 894 (35.0) | 300 (33.6) | 437 (36.4) | 157 (33.9) | |
| Upper Third | 748 (29.3) | 242 (27.1) | 346 (28.8) | 160 (34.6) | |
| Fast Food consumption | | | | | <0.001 |
| <= 2 times/week | 2045 (80.7%) | 687 (78.0) | 1004 (80.4) | 354 (76.5) | |
| 3-4 times/week | 302 (11.9) | 122 (13.8) | 105 (8.8) | 75 (16.2) | |
| >= 5 times/week | 186 (7.3) | 72 (8.2) | 80 (6.7) | 34 (7.3) | |
| Physical Activity | | | | | <0.001 |
| High | 902 (35.3) | 215 (24.1) | 564 (47.0) | 123 (26.6) | |
| Moderate | 747 (29.2) | 373 (41.8) | 267 (22.5) | 107 (23.1) | |
| Low | 906 (35.5) | 304 (34.1) | 369 (30.8) | 233 (50.3) | |

**Notes.**

P-values are for overall difference in mean or proportions across the three studies using analysis of variance for continuous variables and Chi-squared tests for categorical variables.

SD, standard deviation; BP, blood pressure; PCA, principal component analysis; SES, socioeconomic status.

PCA Derived SES category 1 loads highly positive for proportion of population with tertiary education, larger house size and 6 higher population density and strongly negative for poverty and dependency ratio. Higher values indicate higher SES.

PCA Derived SES category 2 loads highly positive for unemployment and population density, and strongly negative for house size. Higher values indicate lower SES.

**Table 2  Summary statistics for participant and community characteristics by blood pressure category.**

| Characteristics | Normal Blood Pressure (BP < 120/80 mmHg) $n = 1807$ | Elevated Blood Pressure or Hypertension (BP ≥ 120/80 mmHg) $n = 749$ | P-value |
|---|---|---|---|
| | **mean (SD)** | **mean (SD)** | |
| Age (years) | 17.9 (2.0) | 18.0 (2.2) | 0.543 |
| Weight (kg) | 62.7 (13.3) | 68.8 (16.7) | <0.001 |
| Height (cm) | 167.2 (9.2) | 169.9 (9.7) | <0.001 |
| Body mass index (kg/m$^2$) | 22.4 (4.7) | 23.9 (5.8) | <0.001 |
| Systolic blood pressure (mmHg) | 107.0 (7.5) | 122.5 (9.6) | <0.001 |
| Diastolic blood pressure (mmHg) | 66.7 (8.2) | 77.4 (10.4) | <0.001 |
| Waist circumference (cm) | 72.5 (10.1) | 76.4 (12.4) | <0.001 |
| Fasting glucose (mmol/l) | 4.0 (1.2) | 4.1 (1.3) | 0.101 |
| PCA-SES Component 1 (z-score) | 0.08 (1.02) | −0.16 (0.94) | <0.001 |
| PCA SES Component 2 (z-score) | 0.02 (1.05) | −0.04 (0.88) | 0.170 |
| | **n (%)** | **n (%)** | |
| Sex | | | <0.001 |
| *Female* | 1112 (61.5) | 334 (44.6) | |
| *Male* | 695 (28.5) | 415 (55.4) | |
| Body Mass Index Category | | | <0.001 |
| *Underweight* | 250 (13.9) | 72 (9.6) | |
| *Normal weight* | 1162 (64.7) | 453 (60.5) | |
| *Overweight* | 272 (15.1) | 128 (17.1) | |
| *Obesity* | 113 (6.3) | 96 (12.8) | |
| Household Possessions Category | | | 0.079 |
| *Lower Third* | 639 (35.4) | 274 (36.6) | |
| *Middle Third* | 616 (34.1) | 278 (37.2) | |
| *Upper Third* | 552 (30.6) | 196 (26.2) | |
| Fast Food consumption | | | 0.058 |
| *<= 2 times/week* | 1428 (79.6) | 617 (83.6) | |
| *3-4 times/week* | 225 (12.5) | 77 (10.4) | |
| *>= 5 times/week* | 142 (7.9) | 44 (6.0) | |
| Physical Activity | | | <0.001 |
| *High* | 586 (32.4) | 316 (42.3) | |
| *Moderate* | 541 (29.9) | 206 (27.5) | |
| *Low* | 680 (37.6) | 226 (30.2) | |

**Notes.**

P-values are for difference in mean or proportions across the blood pressure categories using t-test for continuous variables and Chi-squared tests for categorical variables.

SD, standard deviation; BP, blood pressure; PCA, principal component analysis; SES, socioeconomic status.

PCA Derived SES category 1 loads highly positive for proportion of population with tertiary education, larger house size and higher population density and strongly negative for poverty and dependency ratio. Higher values indicate higher SES

PCA Derived SES category 2 loads highly positive for unemployment and population density, and strongly negative for house size. Higher values indicate lower SES.

Given these interactions with study, in addition to our sex-specific models in our primary results, we show study-specific estimates in Supplemental Information 1.

Bivariate analyses using mixed effects linear regression with systolic BP as the outcome are shown in Table 3. PCA-SES1 showed a statistically significant inverse association with systolic BP among both males and females. A one standard deviation unit increase in PCA-SES1 score was associated with a 1.45 mmHg lower systolic BP among males ($p < 0.001$) and 1.30 mmHg lower systolic BP among females ($p = 0.003$). For PCA-SES2 we report results for linear splines, given the evidence for a non-linear effect mentioned above. Among males, a significant association was seen only for spline 2 ($z$-score - 1 to 0) where a one standard deviation increase in PCA-SES2 (lower SES) was associated with a 2.6 mmHg higher systolic blood pressure. Among females, significant associations were seen for spline 1 ($z$-score $<-1$), spline 2 ($z$-score $-1$ to 0), and spline 4 ($z$-score $>1$). Coefficients were 2.7 for spline 1, $-3.5$ for spline 2 and $-0.5$ at spline 4. Age and BMI were directly associated with systolic BP among both males and females, while fasting glucose showed significant direct association among females only. Household possession categories was inversely associated with systolic BP for the middle third among males and the upper third among females. There was an inverse association with fast food consumption for the 3-4 times/week category compared to the to the $\leq 2$ times/week category for females only. No association was seen for physical activity levels in these models.

Results for bivariate analyses for diastolic BP are shown in Table 4. There were no significant associations between diastolic BP and PCA-SES1, however inverse associations were seen for males at spline 1 and females at spline 3 for PCA-SES2. When elevated BP or hypertension was used as the outcome variable in bivariate analyses (Table S8), only PCA-SES1 was significantly associated with elevated BP or hypertension, and this was seen only among males in the upper third of PCA-SES1 scores (odds ratio 0.65, $p = 0.015$).

Sequential multivariable analyses with systolic BP as the outcome are shown in Table 5. These results were derived from sex-specific mixed effects linear regression models, with cluster variables being community and parish. Model 1 includes adjustments for age, household SES and study. BMI is added to model 2 and glucose, physical activity and diet added in model 3. Among males, PCA-SES1 remained inversely associated with systolic BP in all models. The coefficient was slightly attenuated (i.e., less negative) when adjustments were made in model 1, changing from $-1.45$ ($p < 0.001$) in the unadjusted model to $-1.31$ ($p < 0.001$) in model 1; however, addition of BMI to the model resulted in the magnitude of the coefficient increasing (i.e., becoming more negative) to $-1.49$ ($p < 0.001$). There was no appreciable change in the coefficient when fasting glucose, physical activity and diet were added to the model. Among females, the significant association seen in the bivariate analysis was attenuated and no longer significant when age, household possessions and study were added in model 1. The coefficient reduced in magnitude from $-1.30$ ($p = 0.003$) in the unadjusted model to $-0.46$ ($p = 0.423$) in model 1. The addition of BMI (model 2) and other covariates (model 3) resulted in only very small changes in the coefficient. For PCA-SES2, the association seen at spline 2 in the unadjusted model among males was no longer significant in any of the sequential models, but a significant positive association was seen at spline 4 ($\beta = 0.94$, $p = 0.042$) in model 2 only. Among females, PCA-SES2 was

**Table 3** Coefficients for association with systolic blood pressure for individual and community characteristics for male and female participants in bivariate regression model.

| Characteristics | Males $N = 1110$ β-coefficient (95% CI) | P-value | Females $n = 1446$ β-coefficient (95% CI) | P-value |
|---|---|---|---|---|
| PCA SES Component 1 (z-score) | −1.45 (−1.82, −1.07) | <0.001 | −1.30 (−2.16, −0.44) | 0.003 |
| PCA SES Component 2 (z-score) | | | | |
| Spline 1 (z-score <-1) | 1.73 (−0.14, 3.59) | 0.070 | 2.74 (0.29, 5.20) | 0.028 |
| Spline 2 (z-score -1 to 0) | 2.58 (0.41, 4.76) | 0.020 | 3.51 (2.28, 4.75) | <0.001 |
| Spline 3 (z-score >0 to 1) | 0.55 (−1.98, 3.08) | 0.668 | 0.02 (−1.82, 1.86) | 0.983 |
| Spline 4 (z-score >1) | 0.11 (−0.15, 0.38) | 0.393 | −0.46 (−0.87, −0.05) | 0.027 |
| Age (years) | 1.08 (0.64, 1.54) | <0.001 | 0.40 (0.002, 0.79) | 0.049 |
| Body mass index (z-score) | 2.63 (2.01, 3.26) | <0.001 | 1.87 (1.47, 2.27) | <0.001 |
| Fasting glucose (mmol/l) | 1.24 (−0.15, 2.63) | 0.080 | 0.45 (0.02, 0.88) | 0.042 |
| Household Possessions Category | | | | |
| *Lower Third* | Reference category | | Reference category | |
| *Middle Third* | −1.11 (−2.22, −0.002) | 0.050 | −0.46 (−1.31, 0.39) | 0.285 |
| *Upper Third* | −0.84 (−2.88, 1.20) | 0.418 | −1.59 (−2.60, −0.58) | 0.002 |
| Fast Food consumption | | | | |
| *≤2 times/week* | Reference category | | Reference category | |
| *3–4 times/week* | 0.46 (−0.79, 1.72) | 0.468 | −1.38 (−2.34, −0.42) | 0.005 |
| *≥ 5times/week* | −0.86 (−2.93, 1.21) | 0.418 | −0.43 (−1.63, 0.78) | 0.650 |
| Physical Activity | | | | |
| *High* | Reference category | | Reference category | |
| *Moderate* | 0.24 (−1.01, 1.49) | 0.704 | −0.06 (−1.71, 1.59) | 0.943 |
| *Low* | −1.57 (−3.34, 0.20) | 0.082 | −0.24 (−1.60, 1.13) | 0.735 |

**Notes.**

PCA, Principal Components Analysis; SES, socioeconomic status.

P-values from three level linear mixed effect models in Stata. Level 1—individual study participants; level 2—communities in which participant lived; level 3—parish in which community was situated. Models used multiple imputation to account for missing data and used robust standard errors.

directly associated with systolic BP at spline 2 (z-score -1 to 0) with little change across the models. At this level a one standard deviation increase in PCA-SES2 (lower SES) was associated with a 4.09 mmHg increase in systolic BP in the fully adjusted model. At spline 3 (z-score >0 to 1), higher PCA-SES2 score was associated with lower blood pressure (β = −2.81, p = 0.013 in model 3).

Table 6 shows the sequential multivariable models with diastolic blood pressure as the outcome. There were no significant associations seen for PCA-SES1 in any of the models for males or females. Among males, a significant inverse association was seen at spline 1 for PCA-SES2 (z-score <−1) (β = −4.10, p < 0.001 in the fully adjusted model). There was a significant direct association at spline 2 (z-score −1 to 0) (β = 2.85, p = 0.002). Among females, there was a significant positive association between diastolic BP and PCA-SES2 at spline 1 (β = 1.06, p = 0.023).

Table 7 shows the odds ratio for elevated blood pressure or hypertension derived from sex-specific multi-level logistic regression models adjusted for age category, household SES, study, BMI categories, glucose category, physical activity and fast food consumption. Among males, the odds ratio for elevated blood pressure or hypertension for persons

**Table 4  Coefficients for association with diastolic blood pressure for individual and community characteristics for male and female participants in bivariate regression model.**

| Characteristics | Males $N = 1110$ β-coefficient (95% CI) | P-value | Females $n = 1446$ β-coefficient (95% CI) | P-value |
|---|---|---|---|---|
| PCA SES Component 1 (z-score) | −0.14 (−0.91, 0.63) | 0.720 | 0.15 (−1.22, 1.53) | 0.826 |
| PCA SES Component 2 (z-score) | | | | |
|   Spline 1 (z-score <−1) | −2.73 (−4.10, −1.36) | <0.001 | 0.42 (−1.49, 2.34) | 0.665 |
|   Spline 2 (z-score -1 to 0) | 0.47 (−0.66, 1.61) | 0.412 | −0.57 (−2.23, 1.09) | 0.500 |
|   Spline 3 (z-score >0 to 1) | −1.21 (−2.67, 0.26) | 0.107 | −1.41 (−2.18, −0.64) | <0.001 |
|   Spline 4 (z-score >1) | −0.13 (−0.42, 0.17) | 0.396 | −0.14 (−0.45, 0.18) | 0.399 |
| Age (years) | 0.67 (0.16, 1.17) | 0.010 | 0.17 (−0.12, 0.47) | 0.252 |
| Body mass index (z-score) | 0.55 (−0.46, 1.56) | 0.286 | 0.81 (0.29, 1.32) | 0.002 |
| Fasting glucose (mmol/l) | 0.41 (−0.71, 1.53) | 0.475 | 0.31 (−0.18, 0.79) | 0.215 |
| Household Possessions Category | | | | |
|   *Lower Third* | Reference category | | Reference category | |
|   *Middle Third* | −0.59 (−1.90, 0.71) | 0.372 | 0.40 (−0.38, 1.19) | 0.311 |
|   *Upper Third* | −0.51 (−1.55, 0.54) | 0.343 | −0.27 (−1.71, 1.17) | 0.709 |
| Fast Food consumption | | | | |
|   *≤2 times/week* | Reference category | | Reference category | |
|   *3-4 times/week* | 0.43 (−1.18, 2.04) | 0.602 | 0.05 (−0.93, 1.04) | 0.916 |
|   *≥5 times/week* | 1.26 (−0.56, 3.08) | 0.174 | −0.84 (−2.25, 0.56) | 0.241 |
| Physical Activity | | | | |
|   *High* | Reference category | | Reference category | |
|   *Moderate* | 0.59 (−0.59, 1.77) | 0.329 | −0.15 (−0.86, 0.56) | 0.683 |
|   *Low* | −0.76 (−2.54, 1.01) | 0.399 | −0.97 (−1.85, −0.10) | 0.029 |

**Notes.**

PCA, Principal Components Analysis; SES, socioeconomic statu.

P-values from three level linear mixed effect models in Stata. Level 1—individual study participants; level 2—communities in which participant lived; level 3—parish in which community was situated. Models used multiple imputation to account for missing data and used robust standard errors.

living in communities in the upper third of PCA-SES1 was 0.67 ($p = 0.051$), suggesting that higher SES may be associated with lower odds of hypertension or elevated blood pressure, but did not achieve statistical significance in this study. There were no significant associations among females for thirds of PCA-SES1. Thirds of PCA-SES2 was not associated with elevated blood pressure in males or females.

In order to assess whether our results could have been influenced by the multiple imputation, we re-ran the full models (model 3) with complete cases only (i.e., no imputations) for systolic and diastolic blood pressure. These models are shown in Tables S9 and S10. The associations seen were generally very similar to that seen in the models with multiple imputation but given the smaller numbers (1,019 for males and 1,362 for females) estimates were often less precise. We also assessed whether the observed associations differed according to study by re-running the full final model for each study separately. These results are shown in Tables S11 and S12. While there were some differences in association in the different studies, associations were generally similar in direction to the findings in the pooled analysis, however there was some variation in the magnitude of the

Ferguson et al. (2020), *PeerJ*, DOI 10.7717/peerj.10058

**Table 5** **Multivariable models for association between systolic blood pressure and neighbourhood socioeconomic status for male and female participants.**

| Characteristics | Model 1 | | Model 2 | | Model 3 | |
|---|---|---|---|---|---|---|
| | β (95% CI) | *p*-value | β (95% CI) | *p*-value | β (95% CI) | *p*-value |
| **MALES** | | | | | | |
| PCA SES Component 1 (*z*-score) | −1.31 (−1.80, −0.81) | <0.001 | −1.49 (−2.06, −0.92) | <0.001 | −1.48 (−2.11, −0.84) | <0.001 |
| PCA SES Component 2 (linear splines) | | | | | | |
| Spline 1 (*z*-score <-1) | −0.74 (−4.77, 3.28) | 0.718 | −0.04 (−3.89, 3.81) | 0.984 | 0.10 (−4.01, 3.82) | 0.962 |
| Spline 2 (*z*-score -1 to 0) | 1.45 (−1.87, 4.77) | 0.393 | 0.98 (−2.57, 4.53) | 0.588 | 1.06 (−2.41, 4.52) | 0.550 |
| Spline 3 (*z*-score >0 to 1) | −0.65 (−3.28, 1.97) | 0.627 | −0.68 (−3.48, 2.11) | 0.631 | −0.39 (−3.25, 2.48) | 0.790 |
| Spline 4 (*z*-score >1) | 0.76 (−0.06, 1.57) | 0.069 | 0.94 (0.04, 1.84) | 0.042 | 0.76 (−0.21, 1.73) | 0.127 |
| **FEMALES** | | | | | | |
| PCA SES Component 1 (*z*-score) | −0.46 (−1.58, 0.66) | 0.423 | −0.51 (−1.64, 0.63) | 0.382 | −0.48 (−1.62, 0.66) | 0.410 |
| PCA SES Component 2 (linear splines) | | | | | | |
| Spline 1 (*z*-score <-1) | −0.52 (−2.83, 1.80) | 0.662 | −0.33 (−2.63, 21.97) | 0.778 | −0.43 (−2.74, 1.89) | 0.719 |
| Spline 2 (*z*-score -1 to 0) | 4.02 (1.71, 6.33) | 0.001 | 3.97 (1.36, 6.57) | 0.003 | 4.09 (1.49, 6.69) | 0.002 |
| Spline 3 (*z*-score 0 to 1) | −2.52 (−4.82, −0.22) | 0.032 | −2.84 (−5.02, -065) | 0.011 | −2.81 (−5.04, −0.59) | 0.013 |
| Spline 4 (*z*-score >1) | 0.36 (−1.16, 1.88) | 0.645 | 0.60 (−1.06, 2.27) | 0.478 | 0.63 (−1.05, 2.31) | 0.463 |

**Notes.**

PCA, Principal Components Analysis; SES, socioeconomic status.

Model 1: adjusted for age, household SES, and study

Model 2: adjusted for age, household SES, study, and BMI *z*-score

Model 3: adjusted for age, household SES, study, BMI *z*-score, glucose, physical activity, and fast food consumption

Coefficients for all variables in model three are shown in Table S14.

*P*-values from three level linear mixed effect models in Stata. Level 1—individual study participants; level 2—communities in which participant lived; level 3—parish in which community was situated.

Models used multiple imputation to account for missing data and used robust standard errors.

Ferguson et al. (2020), *PeerJ*, DOI 10.7717/peerj.10058

**Table 6** **Multivariable models for association between diastolic blood pressure and neighbourhood socioeconomic status for male and female participants.**

| Characteristics | Model 1 | | Model 2 | | Model 3 | |
|---|---|---|---|---|---|---|
| | β (95% CI) | p-value | β (95% CI) | *p*-value | β (95% CI) | *p*-value |
| **MALES** | | | | | | |
| PCA SES Component 1 (*z*-score) | −0.09 (−0.76, 0.58) | 0.794 | −0.12 (−0.79, 0.56) | 0.735 | −0.13 (−0.84, 0.59) | 0.730 |
| PCA SES Component 2 (linear splines) | | | | | | |
| Spline 1 (*z*-score <-1) | −4.28 (−6.10, −2.47) | <0.001 | −4.16 (−5.99, −2.32) | <0.001 | −4.10 (−6.11, −2.09) | <0.001 |
| Spline 2 (*z*-score -1 to 0) | 2.80 (0.86, 4.75) | 0.005 | 2.73 0.80, 4.66) | 0.006 | 2.85 (1.05, 4.66) | 0.002 |
| Spline 3 (*z*-score >0 to 1) | −2.58 (−4.71, −0.43) | 0.018 | −2.59 (−4.72, −0.46) | 0.017 | −2.36 (−4.88, 0.17) | 0.067 |
| Spline 4 (*z*-score >1) | 0.76 (0.02, 1.50) | 0.045 | 0.79 (0.06, 1.52) | 0.033 | 0.60 (−0.25, 1.44) | 0.167 |
| **FEMALES** | | | | | | |
| PCA SES Component 1 (*z*-score) | 0.50 (−0.68, 1.68) | 0.406 | 0.48 (−0.69, 1.65) | 0.421 | 0.50 (−0.65, 1.65) | 0.395 |
| PCA SES Component 2 (linear splines) | | | | | | |
| Spline 1 (*z*-score <-1) | 1.11 (0.22, 2.0) | 0.014 | 1.18 (0.28, 2.08) | 0.010 | 1.06 (0.15, 1.98) | 0.023 |
| Spline 2 (*z*-score -1 to 0) | 0.19 (−1.68, 2.07) | 0.843 | 0.17 (−1.66, 2.01) | 0.854 | 0.27 (−1.58, 2.11) | 0.777 |
| Spline 3 (*z*-score 0 to 1) | −1.47 (−3.79, 0.85) | 0.215 | −1.61 (−3.82, 0.59) | 0.153 | −1.49 (−3.73, 0.76) | 0.194 |
| Spline 4 (*z*-score >1) | 0.39 (−0.69, 1.47) | 0.480 | 0.49 (−0.56, 1.54) | 0.358 | 0.46 (−0.61, 1.54) | 0.397 |

**Notes.**

PCA, Principal Components Analysis; SES, socioeconomic status.

Model 1: adjusted for age, household SES, and study

Model 2: adjusted for age, household SES, study, and BMI *z*-score

Model 3: adjusted for age, household SES, study, BMI *z*-score, glucose, physical activity, and fast food consumption

Coefficients for all variables in model three are shown in Table S13.

*P*-values from three level linear mixed effect models in Stata. Level 1—individual study participants; level 2—communities in which participant lived; level 3—parish in which community was situated.

Models used multiple imputation to account for missing data and used robust standard errors.

**Table 7  Multivariable model for association between elevated blood pressure or hypertension and covariates for male and female participants.**

| Characteristics | Males Odds ratio (95% CI) | | Females Odds ratio (95% CI) | |
| --- | --- | --- | --- | --- |
| PCA SES Component 1 (Thirds) | | | | |
| *Lower Third* | Reference category | | Reference category | |
| *Middle Third* | 0.81 (0.58, 1.13) | 0.201 | 0.94 (0.65, 1.37) | 0.742 |
| *Upper Third* | 0.67 (0.44, 1.00) | 0.051 | 1.08 (0.67, 1.74) | 0.760 |
| PCA SES Component 1 (Thirds) | | | | |
| *Lower Third* | Reference category | | Reference category | |
| *Middle Third* | 1.02 (0.70, 1.49) | 0.910 | 1.21 (0.79, 1.86) | 0.387 |
| *Upper Third* | 0.85 (0.59, 1.24) | 0.400 | 1.10 (0.70, 1.74) | 0.684 |

**Notes.**

PCA, Principal Components Analysis; SES, socioeconomic status; JHLS-II, Jamaica Health and Lifestyle Survey 2007-2008.

Analyses included 1,108 males and 1438 females.

Models adjusted for age (categories), household SES, study, body mass index categories, high glucose (upper quintile), physical activity level and fast food consumption (surrogate for high salt diet).

Coefficients for all variables in model three are shown in Table S15.

*P*-values from three level logistic mixed effect models in Stata. Level 1—individual study participants; level 2—communities in which participant lived; level 3—parish in which community was situated. Models used multiple imputation to account for missing data.

estimates; additionally, estimates were again imprecise with wide confidence intervals and often not statistically significant.

# DISCUSSION

We have shown from these analyses that lower community SES is associated with higher systolic BP among youth in Jamaica, particularly among males. This association persisted after adjusting for possible confounding by age and household SES and after accounting for the effects of BMI, fasting glucose, physical activity and diet (fast food consumption). Among females, the association with the first PCA-SES component was no longer significant after adjusting for confounders. Associations for the second PCA-SES component demonstrated non-linear effects. Diastolic BP also showed some non-linear associations for the second PCA-SES variable. However, while lower community SES may also be associated with higher odds of elevated blood pressure or hypertension among males, this did not achieve statistical significance.

The findings from this study adds to the body of literature showing an important contribution of neighbourhood conditions to health and underscores the importance of including the neighbourhood context when evaluating the social determinants of health. It also highlights that the relationship between neighbourhood SES and blood pressure may be complex, with both linear and non-linear effects, thus necessitating a nuanced approach to our interpretation of these associations. A relationship between neighbourhood SES and blood pressure in adults is fairly well established, with several studies showing that lower neighbourhood SES was associated with higher BP (*Coulon et al., 2016*; *Fan et al., 2015*; *Liu et al., 2013*; *Morenoff et al., 2007*; *Riva, Larsen & Bjerregaard, 2016*; *Sprung et al., 2019*). Additionally, Cho and colleagues found that lower neighbourhood SES was associated with increased all-cause mortality among persons with newly diagnosed hypertension in Korea (*Cho et al., 2016*). The relationship between neighbourhood SES and BP among

adolescents and young adults is less clear with some studies showing an inverse association and others no significant association. Murakami and colleagues found that neighbourhood socioeconomic disadvantage was associated with higher blood pressure among Japanese dietary students at age 18–22 years, with those in the highest quartile of neighbourhood socioeconomic disadvantage having 3 mmHg higher SBP (*Murakami et al., 2010*). In another study, McGrath and colleagues found that neighbourhood income, measured as percent at or below the poverty line, was significantly associated with ambulatory systolic BP among 14-year old adolescents in Pittsburgh (*McGrath, Matthews & Brady, 2006*). In that study, each 1% increase in the percentage of community members at or below the poverty line was associated a 0.29 mmHg increase in systolic BP. Hofelmann and colleagues, in a study conducted in Brazil among persons 20–29 years old, found that area level income was inversely associated with both SBP and hypertension; being in the upper third of community income was associated with 5.7 mmHg lower SBP (*Hofelmann et al., 2012*). Data from the Birth to Twenty Cohort in South Africa also found significant associations with a number of neighbourhood SES indices in bivariate analyses at age 16 years, however only one of these remained statistically significant in the fully adjusted model (*Griffiths et al., 2012*). Kwok and colleagues found that relative household income, but not neighbourhood income, was inversely associated with blood pressure in Chinese adolescents at age 13 (*Kwok et al., 2015*); while Chen and Paterson found no significant association between neighbourhood SES and blood pressure (*Chen & Paterson, 2006*). In the Hong Kong Growth Study of children 6-19 years, Ip and colleagues reported a significant association between maternal education and hypertension, but no significant association between community income and hypertension (*Ip et al., 2016*). Community income was however significantly associated with obesity. The study did not report associations with SBP or DBP as continuous variables; such analyses may have been more likely to show an association given the higher statistical power associated with continuous variable (*Altman & Royston, 2006*).

In this study, the association between neighbourhood SES and diastolic BP was seen only for the second PCA-SES variable and showed non-linear effects. We have previously reported weaker associations between diastolic BP and socioeconomic variables in a longitudinal analysis from the 1986 Birth Cohort (*Ferguson et al., 2015*). Additionally, McGrath and colleagues found that while neighbourhood SES was associated with systolic BP among adolescents in Pittsburgh, there was no significant association with diastolic BP (*McGrath, Matthews & Brady, 2006*). On the other hand, Riva and colleagues in a study from Greenland found that while neighbourhood SES was associated with both systolic BP and diastolic BP, the association was inverse U-shaped (*Riva, Larsen & Bjerregaard, 2016*), while Coulon also found significant associations for both systolic BP and diastolic BP among African-American adults (*Coulon et al., 2016*). Taken together these studies suggest that there is some variability in the relationship between neighbourhood SES and diastolic BP and that unraveling these relationships will require further studies.

The finding of sex differences in the association between community SES and health have been previously reported in studies from Jamaica (*Cunningham-Myrie et al., 2015*; *Mullings et al., 2013*). Cunningham-Myrie and colleagues (*Cunningham-Myrie et al., 2015*)

found significant sex interaction in the relationship between neighbourhood infrastructure score and the prevalence of obesity, while Mullings and colleagues (*Mullings et al., 2013*) found that significant sex differences in the relationship between prevalent depressive symptoms and neighbourhood infrastructure and settlement pattern. In a study from Greenland, Riva and colleagues found that the associations between neighbourhood SES and blood pressure appeared to be stronger in men compared to women (*Riva, Larsen & Bjerregaard, 2016*). These findings are consistent with several other studies from our group that have shown sex-differences in the relationship between individual or household level socioeconomic status and health outcomes (*Ferguson et al., 2010a*; *Ferguson et al., 2015*; *Ferguson et al., 2018*; *Ferguson et al., 2017*; *Ferguson et al., 2010b*). This suggests that sex may be an important effect modifier in the relationship between social factors and health and therefore should be explored in studies evaluating these relationships. The associations found in this study could be interpreted to suggest that education and wealth (high loadings in PCA-SES1) may be protective for men, whereas poverty and crowding (as represented by unemployment and population density) may have important effects for women. This is supported by the findings of Mullings and colleagues who found that for depressive symptoms in Jamaica, poor community infrastructure was associated with increased risk among males, whereas, among females, residing in informal unplanned (usually crowded) communities was associated with increased risk (*Mullings et al., 2013*). It is possible that increased stress from living in poorer, tension-prone environments, as well as higher levels of alcohol and tobacco use may contribute to the higher blood pressure from lower SES for males, whereas household stressors related to crowding, lack of financial resources contribute to higher blood pressure in females. These hypotheses would need to be further explored in future studies.

The mechanisms underlying the relationship between neighbourhood SES and blood pressure has not yet been fully elucidated. Broadly speaking, neighbourhood SES may influence health through compositional effects (based on aggregate effect of individual risk factors), contextual effects (group level effects beyond those due to risk factors) or environmental effects (*Diez Roux & Mair, 2010*). This involves the process of embodiment where social and psychological factors in one's environment result in biological changes and are ultimately expressed as biological characteristics (*Krieger, 2005*). With regards to blood pressure, it has been suggested that persistent vigilance in a setting of neighbourhood disorder or poor neighbourhood social circumstances may induce changes in vascular reactivity which results in higher blood pressure (*Ewart et al., 2004*). Another study found that agonistic striving (induced by challenges to interpersonal influence and control) and subordination (due to social denigration, rejection and mistreatment) are potential mechanisms through which neighbourhood stress may result in hypertension (*Ewart, Elder & Smyth, 2014*). Other studies have found associations between adverse neighbourhood circumstances and reduced nocturnal blood pressure dipping (*Euteneuer et al., 2014*; *Mellman et al., 2015*). Taken together these studies suggest that neural regulation of blood pressure may be altered by psychosocial stress induced by living in neighbourhoods with adverse social and economic conditions.

### Strengths and limitations

This study had several strengths, including the fact that individual participant data for all three studies were available and therefore we were able perform pooled analyses in the study. The similarities in the source population, age range, and timing for each study the studies facilitated pooled analysis. While there was some evidence of heterogeneity between studies, this was seen only for specific splines of the second PCA-SES variable and therefore would have little effect on our conclusions. The studies in general had good response rates, and two were from nationally representative samples, so that the findings can be considered as generalizable to the Jamaican population. Another strength of this study is the use of PCA to derive socioeconomic status scores thus taking into consideration several SES variables rather than only a few select community characteristics.

Limitations of this study include the cross-sectional design which limit our ability to make causal inferences. However, given the findings from other cross-sectional and longitudinal studies cited above, our findings add to the literature on this subject and strengthens the consistency of the associations reported. The study was also limited by the fact that we had missing data for some variables, however we were able to account for this by using multiple imputation analyses, assuming that data were missing at random. We assessed the effect of imputation on the findings by performing complete case analyses, which showed findings which were very similar to those found when imputation was used, thus increasing our confidence in the findings reported. Another potential limitation was the possibility of the same individual being included in two (or three) of the studies. While some communities were included in more than one study, we thought that the probability of the same individual being selected was low, given the differences in inclusion criteria for each study and the fact that households were selected based on a random starting point in the national surveys. It is therefore unlikely that such duplication would have had any meaningful effect on our estimates. Finally, we were unable to use survey weights to adjust for probability of selection of participants in communities as these data were available for only one of the three studies used; however, given the fact that we were more focused on the associations between neighbourhood SES and blood pressure rather than obtaining estimates of population prevalence, we believe that this would have very little effect on our findings. This position is supported by Carle who found that there were only minimal differences, which did not change inferences, when comparing weighted vs. unweighted analyses for multilevel models using survey data (*Carle, 2009*).

Overall, we believe that the findings in this paper will make an important addition to the literature on the relationship between neighbourhoods and health, particularly in light of its focus on adolescents and young adults in an African origin developing country context. This will serve as a catalyst for further research in this area and will help to inform our strategy for reducing the adverse public health implications of hypertension.

## CONCLUSION

We have demonstrated that neighbourhood SES was inversely associated with BP among Jamaican youth and that there were sex differences in these associations. Findings from this

study suggest that the neighbourhood context may be an important factor in the aetiology of hypertension and that interventions to address the growing public health challenges resulting from hypertension should include evaluation of neighbourhoods. Further research in Jamaica and similar populations is warranted to improve our understanding of the aetiology of elevated blood pressure and to help in the development of appropriate interventions.

## ACKNOWLEDGEMENTS

The authors acknowledge the contribution of the project staff (nurses, laboratory personnel, administrative staff, and project assistants) and the individual study participants for their contribution to the project. The authors also acknowledge the contribution of the Mona Geo-Informatics Institute for its contribution with regards to the community socioeconomic characteristics data.

### Funding

The studies included in this paper were funded by grants from the National Health Fund (Jamaica), the Caribbean Health Research Council, the Caribbean Cardiac Society, the Culture Health Arts Sports and Education Fund (Jamaica), the University Hospital of the West Indies, and the U.S. Agency for International Development (USAID). The funders had no role in study design, data collection and analysis, decision to publish, or preparation of the manuscript.

### Grant Disclosures

The following grant information was disclosed by the authors:
National Health Fund (Jamaica).
Caribbean Health Research Council.
Caribbean Cardiac Society.
Culture Health Arts Sports and Education Fund (Jamaica).
University Hospital of the West Indies.
U.S. Agency for International Development (USAID).

### Competing Interests

The authors declare there are no competing interests.

### Author Contributions

- Trevor S. Ferguson conceived and designed the experiments, performed the experiments, analyzed the data, prepared figures and/or tables, authored or reviewed drafts of the paper, and approved the final draft.
- Novie O.M. Younger-Coleman conceived and designed the experiments, performed the experiments, analyzed the data, authored or reviewed drafts of the paper, and approved the final draft.

- Jasneth Mullings, Damian Francis and Rainford Wilks conceived and designed the experiments, authored or reviewed drafts of the paper, and approved the final draft.
- Lisa-Gaye Greene performed the experiments, prepared figures and/or tables, authored or reviewed drafts of the paper, and approved the final draft.
- Parris Lyew-Ayee performed the experiments, authored or reviewed drafts of the paper, and approved the final draft.

## Human Ethics

The following information was supplied relating to ethical approvals (i.e., approving body and any reference numbers):

The University of the West Indies Ethics Committee approved this research (proposals: ECP 173, ECP 161, ECP 71).

## Data Availability

The data are available as Supplementary Files.

## Supplemental Information

Supplemental information for this article can be found online at http://dx.doi.org/10.7717/peerj.10058#supplemental-information.

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
