# Peer review of "Neighbourhood socioeconomic characteristics and blood pressure among Jamaican youth: a pooled analysis of data from observational studies"

_PeerJ, doi:10.7717/peerj.10058_

## Round 0.1 · original submission · Major Revisions

The three reviewers have collectively made some positive comments about your manuscript, and I agree that this work has the potential to make a useful addition to the literature. They have also raised some important points and asked some questions that I think would also be asked by other readers, and so I will ask you to respond to each of their points separately, indicating in each case how you have changed the manuscript in response or why you do not believe any such changes are needed. I think that all the comments from all of the reviewers are worthy of response, and I will add some of mine below, linking these to those from the reviewers as appropriate. I look forward to seeing a revised version of your manuscript.

The motivation for the present study (and I personally think it probably is justified) needs to be made clearer in the introduction (see Reviewer #3’s comments). The reader should be able to see the research gap before the final “paragraph” in this section (Lines 85–91 here). The lack of research in Jamaica (Lines 83–84) is important but does not seem to me to be sufficient argument without some reason to consider it possible that the association(s) in Jamaica might differ from those in other settings.

The operationalisation of 120/80 as high BP needs to be explained (see Reviewer #1’s comment).

Related to this, the classification of the communities’ SES should be explained, including why a new approach was used (see Reviewer #3’s comment). The use of principal components does raise the question of why this was done if individual variables from the PCA are then referred to (see Reviewer #3’s comment) and this should be made clear. It is common enough for sets of variables such as these to be used to construct some form of deprivation index, and normally in this case, the index would be used rather than its constituents. Can you explain why you look at both the PC scores and the constituents of these?

An essential aspect of a manuscript like this is making sure that the reader understands the underlying causal model that is informing your statistical analyses. While you cannot conclude that there are causal associations (see Reviewer #3’s comment here, and you acknowledge this limitation in the discussion), this model will help readers to appreciate how you see potential confounding and effect modification, as well as helping to identify when adjustments might be inappropriate (e.g. colliders or intermediate variables). Reviewer #3 makes some comments about this and I will add that I would be concerned about BMI, waist circumference, and fasting glucose, for example, as potential mediators between SES and BP (e.g. if the association was in part driven by some of: poor quality diet, lack of physical activity, disturbed sleep, stress, poor access to healthcare services, or some of the other possible drivers identified by Reviewer #2, it seems plausible that the association between SES and BP is at least partially mediated through these variables and so you may be over-adjusting if you are interested in, as I assume to be the case, the total and not just the direct effect). If possible, providing a graphical representation of the underlying causal model would help greatly here, ideally a DAG with the relationships, and the lack of other relationships, justified. A less formal representation would still help readers to see how your expectations around the relationships between the variables has influenced your modelling strategy.

While stratification by sex is potentially fine (it avoids the need to include or decide how to include interactions for all covariates, which removes the efficacy gains from using a combined data set), readers will want to know if there is evidence of sex as an effect modifier for each model (rather than the blanket statement on Line 181) and it is important that the variables included in these three (SBP, DBP, and high BP) models are clear here along with any other necessary details for a reader to replicate your analyses using your data. See Reviewer #2’s comment and consider adding interaction p-values to address this question. Reviewer #3 asks an important question about interpreting non-significance within a strata, and addressing this will be much easier with these interaction p-values.

I appreciate you providing your sample size calculations, but more information is needed for this to be replicable. Some of this is reasonably standard (the power and level of significance) but I’m not sure how you powered for the gradient as, assuming 80% and two-sided 0.05, none of my guesses match your figure of 384 (which appears to be a total and so I would have thought would have a multiple of 3 if the SES communities were the same size). It is not clear where your design effect of 1.72 (Line 158) came from. Can you please ensure that you have sufficient information provided to ensure that readers could replicate your calculations themselves (the calculations for the design effect could be included in a supplement if they are too long to include here)? The 4mmHg difference used here seems to have been empirically determined, whereas it is the relevance of this effect that should be used (what is the smallest difference that would be of clinical or public health interest?) Similar comments apply to Lines 163–172, where you should again focus on the power to detect a particular effect size. Note that the point on Lines 169–172 would not necessarily apply with design effects to accommodate the studies themselves.

More explanation of the statistical models is needed (see also Reviewer #3’s comment about crossed-effects models). Related to this, more information about the statistical model diagnostics is needed (see Reviewer #1’s comments about outliers as part of this). I’m not able to see how the xt commands (Line 181) would assist with descriptive statistics.

One part that I wondered was why you collapsed the SES PC scores into thirds by using tertiles. I appreciate that non-linearities are plausible, but these could have been addressed in other ways (e.g. splines, the addition of a quadratic term, fractional polynomials, etc.) without sacrificing statistical power to the extent you have by using tertiles. The effects on Lines 41, are not drastically non-linear in any case (-4.1, -2.4, and 0).

Imputation models are just as complicated as analytical models and more information is also needed here (see Reviewer #1 and #3’s comments). As Reviewer #3 notes, the complete case analysis results should be included as a supplement. The impacts of MCAR, MAR, and NMAR data should be discussed.

I think more interpretation of clinical/public health significance, both for non-statistically significant and statistically significant results, is needed. This should help to address Reviewer #3’s question about DBP. Related to this, more effect sizes and CIs in the text would help the reader to appreciate this important aspect of your findings.

The way the three data sets were combined warrants more discussion (see Reviewers #1, #2, and #3’s comments). This includes adjusting for the data set, which would be expected in an IPD meta-analysis and seems sensible here also. Based on Lines 204–207, it sounds as if you have done this with the study level random effect, but this is based on only three levels of this factor, which is fewer than generally recommended (see specific comment below). Some aspects of the statistical modelling here were not quite clear to me and I (and I suspect some other readers) would appreciate the Stata code being also included as a supplement to ensure that any potential uncertainties in the manuscript can be resolved by looking at the actual code. I also think many readers would wonder about whether or not associations differed between these studies and you could use Reviewer #2’s suggestion about looking at each data source separately, perhaps leading to useful supplementary tables, as one option here (I appreciate that the manuscript is already quite long). How confident are you that there could not be any overlap between the participants in each study (the ages and years suggest that this is possible)? The description on Line 205 doesn’t make it clear if this was known (and so participants were crossed with studies) but each participant number appears to be distinct and so I’m assuming they were all treated as independent.

Reviewers #2 and #3 make some useful suggestions about positioning your work within the literature and it should be possible to extend the discussion by considering their suggestions.

An additional table with the neighbourhood variables would be a valuable addition (see Reviewer #3’s comment). I think a table showing the study characteristics for each of the three data sources, including these neighbourhood variables, would be extremely useful to readers to complement the brief descriptions in the text. Some of the existing tables and figures could also be removed or moved to supplementary material (See Reviewer #3’s comments). The graphs could be improved (see Reviewer #2’s comments), perhaps including removing Stata’s default grid lines and shading.

Given the age range of participants, BMI values should be replaced by BMI z-scores to make these comparable (a BMI of 18.4 for a 15 year old is not quite the same as a BMI of 18.4 for a 24 year old).

Note also that the BMI categories do not completely agree with the descriptions in the supplementary document Supplementary_file_2B_Variable_Names_and_Codes.docx (this says, as expected, that “4 >=30 kg/m.sq.”). However, the minimum and maximum values do not match up with the BMI categories.

Summary for variables: bmi
by categories of: bmicatn4 (bmicatn4)

bmicatn4 | min max
---------+--------------------
1 | 18.50477 24.99976
2 | 13.92261 18.48146
3 | 25.0052 29.99242
4 | 22.99563 60.66777
---------+--------------------
Total | 13.92261 60.66777
* * *
There are two values under 30 (23.0 and 28.6, participants 1567 and 1823 respectively) included in the obese category. Both of these participants have values for highbp_120_80 and so will be included in analyses.

Specific comments:

Line 36: While notation using ± usually indicates mean ± SD, this is not the only interpretation, and so it should be made clear at first use in the abstract, and again in the manuscript, how the reader should interpret this (as you do in Table 1). See also Line 238.

Line 42: Perhaps consider “(BOTH compared to lower THIRD)” here. Note that tertiles are the percentiles and not the categories themselves. See https://en.wikipedia.org/wiki/Quantile for details and consider using “third(s)” rather than “tertile(s)” throughout the manuscript.

Line 55: I don’t think you want the comma after “particular” here.

Lines 85–91: The meaning of these statements hinges on what covariates are included in the models (an aim intending only unadjusted analyses and one intending to adjust for all important confounders, or one using an instrumental variable approach, and I’m not claiming that there is a suitable instrument here, would be addressing quite different questions) and I think readers will benefit from knowing what variables (even if this is described only briefly) are intended to be adjusted for when addressing these aims.

Line 96: I suggest either “…youth aged 15–24…” or “…youth aged between 15 and 24…”. See also Line 120.

Lines 98–101: Perhaps order these here (and elsewhere) in chronological order of starting data collection?

Lines 102–103 and 104–106: Please provide ethics approval numbers here.

Line 106: Please clarify whether this consent was for the original and/or pooled studies (I’m assuming the former).

Lines 142, 144–146: Please include manufacturers and models for the devices (stadiometers, scales, and tape) used if possible.

Line 151: I suggest that “calculations” are “performed” rather than “estimated” (although values can be “estimated”).

Line 155: I don’t think you need to give the variance alongside the SD.

Line 161: I don’t think “therefore” works here (it doesn’t link the first and second clauses of this particular sentence).

Lines 181–183: This is perhaps getting into results, although it is certainly easier to present here. Consider this simply a note and I’ll leave any changes based on it up to you. I do think that readers would want to see p-values if it remained as is though (including for the DBP and the hypertension models).

Line 183: As a stylistic suggestion, “univariable” might be better than “bivariate” here given you then talk about “multivariable” models.

Lines 188–189: Do you mean “a combination” rather than “the combination” as there is no requirement for there to be only one such combination.

Line 189: Do you mean “p<0.05” here? (I appreciate that your intention is fairly obvious, but there is a test statistic for Bartlett’s test and the potential for ambiguity, however small, should be resolved.)

Lines 190–192: While I understand the empirical approach used here, do you feel that this set of variables also achieves content (and perhaps also face) validity? By this, I mean that the selection cannot be purely automated as essential variables could then be omitted, and I would guess that you judged this set as reasonable and complete in some way?

Lines 194–195: Horn’s method is these days regarded as preferable to the Kaiser rule. Can you justify why you have used this older approach? Note that “eigenvalue” is normally a single word (see here and elsewhere), and the same applies to “eigenvector”.

Line 196: Perhaps “conducted” rather than “computed” here?

Line 198: “effects” suggests a causal association, which is more than can be concluded from this design (unless you feel you can argue that all confounding is accounted for).

Lines 198–207: I think that you need to be very explicit about the fixed and the random effects here and also the estimation procedure for each of these sets of models (e.g., REML or ML for the linear mixed models?)

Line 206: The general wisdom would be to use random effects only when the levels of that factor number at least five. Can you please justify your use of a random effect with only three levels? There is a brief discussion of this in the excellent (and I am biased as I was the editor for this paper) https://dx.doi.org/10.7717%2Fpeerj.4794 (specifically the section “Considerations when Fitting Random Effects”). A quick look at a random intercepts only model for hypertension reveals convergence issues with all four levels of the data included (i.e. three random effects), a result that also occurred when the two SES categorical variables were included. I may be doing things differently to what you did of course and providing your Stata code will help readers (including myself) understand exactly what was done, and allow us to consider such matters more carefully.

Lines 208–210: I think it would be useful to provide a little more information here, including what the (regression?) models were for each variable with missing data and how many burn-in iterations were used?

Line 211: “fewer” as this is a countable noun.

Line 219: 10 is a very small number of imputations to use these days. This, or even fewer, was common when computers were much slower, but there is literature recommending more (say 100 imputations, see https://doi.org/10.1007/s11121-007-0070-9). Can you justify this choice?

Lines 222–229: These model selection steps bias the final effect sizes and p-values for these variables. Can you justify this approach? I would have suggested either simply retaining all variables (to avoid the model selection issue entirely) or using Sander Greenland’s approach to identifying confounders as variables that cause the effect size (or the CI limits of the estimate) for the exposures of interest to change in a meaningful way rather than your seemingly Hosmer-Lemeshow-like screening approach as you are primarily interested in the associations involving the SES PCA scores. I’m happy for you to use this approach as long as you can justify it.

Lines 228–229: This seems slightly fraught as the better model (as measured by AIC) will depend on whether or not transformations of the PC scores were investigated to address any non-linearities. Thoughts?

Line 236: I would like to see the addition of a description of model diagnostics around here (see comment above) that you used to reassure the reader that the mathematical assumptions implied by the regression models were sufficiently well satisfied.

Lines 289–290: Missing “with” here I think.

Lines 310–311: I don’t think you can justify this assumption, although I agree that it is unlikely that suppression effects apply here so that the multivariable model would reveal new evidence of associations. Can you either add in these models or justify this step more carefully?

Line 314: I assume that this is based on BLUPs? Lines 231–232 in the methods were not entirely clear. Again, this is a situation where the Stata code would be extremely useful (“we used post-estimation commands to” isn’t clear in this instance at least).

Lines 314–319: To be honest, I don’t find these figures particularly useful. You could consider moving them to the supplementary material. See Reviewer #3’s comment on this also.

Lines 324–325: Again, it would be useful to present p-values or at least acknowledge that the interaction terms were statistically significant when this was the case (just SBP). Or perhaps do this for the multivariable models instead?

Line 326: Missing comma between “BMI” and “fasting”.

Lines 327–329: Again, p-values for interactions would help readers to appreciate the meaning of these (evidence for one sex and an absence for the other not necessarily implying evidence for effect modification; and evidence or an absence of evidence for both sexes the same not necessarily implying a lack of evidence for effect modification).

Lines 354–355: This is a more definite statement than I would think could be justified simply by the sample size.

Line 359: Again, n=315 is not inherently too small (nor is it inherently large enough). I think you would have a better argument here and for the above if you referred directly to the widths of their CIs.

Lines 362–264: Again, I think the stronger argument would be to look at the lower and upper CI limits and see if these are of clinical/public health relevance.

Line 365: “more likely to show an association” (not “and”).

Lines 365–366: I think you’ll need to explain your point here more for me. Misclassification only applies to categories and not to continuous measurements (where it becomes measurement error).

Lines 406–407: I think that this point would be stronger if you contrasted it with other studies that did not accommodate clustering. By itself, this is (or at least should be) normal and expected practice, but you could certainly argue that your results should be trusted more than those from studies that didn’t do so (assuming all other things are equal).

Line 412: But note that this will not accommodate NMAR data.

Line 414: Do you mean “…as these data were not available; FOR ALL STUDIES;” as the variable pstratwt is available for one of the studies?

Lines 414–417: I suggest adding some references for this point (I agree with you, but the point should be supported by the literature).

Lines 417–419: I don’t agree here. The use of random effects will assist with accommodating design effects due to clustering within PSUs, but it won’t address weights based on selection and response probabilities or (beneficial) design effects from stratification (if any of these studies did this). Thoughts?

Line 423: This seems to be your only use of “gender” (the social construct) rather than “sex” (the biological characteristic). I assume it should also be “sex”?

Table 1: Some of these variables are clearly positively skewed (e.g. population, population density, poverty, murder rate, and tertiary education) and it would be worth thinking about whether other measures of location (e.g. geometric means or medians) and spread (e.g. geometric SDs or IQRs) would be more appropriate here (and all other variables should also be checked for skew). The same point also applies to Table 3. I’d prefer to see an additional column for p-values rather than an asterisk system, and the specific test used in each case should also be clear from the table notes.

This point about p-value and their source also applies to Tables 2, 3, 4, 6, 7, 8, 9, and 10 (for the last 4 of these, a separate column wouldn’t be needed). I appreciate that the addition of p-values will make the tables more cluttered but readers, especially those interested in planning their own research in this area, will want to be able to distinguish p=0.051 versus p=0.510 and, to a lesser extent, p=0.010 versus p=0.049.

Table 5: Is there a reason for not presenting the p-values for the overall tests across the three groups or tests of (linear and perhaps quadratic if appropriate) trends here? The source of the p-values should be included in the table notes.

Tables 7 and 8: For the PCA scores, it might be useful to report a standardised effect size for these to assist with interpretation.

Supplementary tables: Similar comments as above regarding p-values and their sources.

·

Basic reporting

see below

Experimental design

see below

Validity of the findings

see below

Additional comments

This is an important finding. Neighborhood factors are still understudied and although cross-sectional, this work provides some important links. I have only minor points:
In line 88 you claim that BP ≥ 120/80 is “elevated blood pressure”. This low threshold (instead of a 140 SBP threshold for example) should be explained by studies which provide evidence for the clinical relevance and a better prognostic value than other cut-off values.
Please provide more information about missing data and the detailed multiple imputation process and also about the combining procedure.
Please provide information whether you screen for uni- and multivariate outliers which may bias result.

Reviewer 2 ·

Basic reporting

This is an exceptionally well-written and well-analyzed paper on the association between neighborhood SES and blood pressure in Jamaican youth. The data are a bit old but this is one of the very few papers ever published on this topic from that part of the world. All things considered, my evaluation is very positive.

Experimental design

This is an observational study. Existing data were used. Two different data sets were combined.

Validity of the findings

I would like to see replication of the findings in each data set. As the two data sets are different in most characteristics , I am afraid that some of the results may be arbitrary and due to the combining two data sets. Thus, we need to see a paragraph on whether the authors could find identical results in each data set or not.

Additional comments

Overall, the paper is very strong. There are a few comments to enhance the paper:
There is a literature in the US and Europe which is above what Ana Diez roux has done. Diez Roux has done very strong work but there are also other part of the literature which is missing in this paper. Violence victimization and fear of unsafe neighborhood as well as physical and social aspects of neighborhood all correlate with exercise, BMI, and cardio-metabolic risk, which can inform the arguments made in the discussion of this paper. Another part of the literature that is not engaged here is the density of healthy food / fast-food stores and tobacco and liquor stores, all having implications for blood pressure of the individuals and populations.
The results are all based on gender/sex stratification. This should be justified well in the background and methods and also be reflected in the title of the paper. What happens to the models if you run sex/gender as interaction rather than strata?
The graphs are all one type, and may not best present the data. Currently it is difficult to see the gender differences in 1 graph. Please consider other alternatives that may better show some of the interactions between gender/sex and the associations of interest.
Finally, two distinct data sets were combined. Why a variable reflecting the data set was not controlled in the modeling? A nominal instrumental variable that reflects the study / year.

Reviewer 3 ·

Basic reporting

The manuscript ‘Neighbourhood socioeconomic characteristics and blood pressure among Jamaican youth: a pooled analysis of data from observational studies’ covers an interesting and relevant topic, for which there is limited evidence in Jamaica.
The manuscript is clear, well-written, and very easy to follow. Yet, there are a number of issues that should be addressed before publication:

ABSTRACT:
L44: ‘associated with higher systolic BP (β +2.6 [1.0, 4.2], p<0.01’ – Please remove the plus sign (‘+’)

INTRODUCTION
L75: The introduction lacks the rationale for the study. The authors refer that their study is needed because there is no evidence about Jamaica. Yet, it is unclear what makes Jamaica different from other contexts where this topic has been extensively addressed. Please add a few sentences about the added value and novelty of this study. Finally, the focus of the study on youth population should also be justified.
L74: I also recommend the authors to refer that neighbourhood deprivation has been associated not only with blood pressure but with a series of other biological markers. Please check the papers by Ribeiro et al (https://www.ncbi.nlm.nih.gov/pubmed/29843403, https://www.ncbi.nlm.nih.gov/pubmed/31217447)
L63: Finally, although the authors briefly describe how neighbourhood SES may influence health, this section is too brief. I strongly recommend the authors to describe into more depth the mechanisms (psychosocial, material, behavioral) neighbourhood deprivation may influence blood pressure specifically.

METHODS
L113: ‘data on the built environment…’ - since this study does not rely on build environment data, this sentence can be removed.
L130 onwards: Many details on the neighbourhood-level variables and its geographical scale are missing. Please add a supplementary table describing all the neighbourhood level variables that were used, together with information on the scale (type of administrative unit, and corresponding average area and population size), date of data collection and data source. Also, they should explain why they selected that specific set of community-level socioeconomic variables. Previous studies on deprivation index construction tend to make the first selection based on data availability and based on theoretical constructs (please see: https://jech.bmj.com/content/70/5/493)
L138: The authors mention their territorial unit of analysis is the community-level. Yet, this is not enough as the term ‘community’ means different things across different countries and regions. Please explain how many communities exist in Jamaica, provide data on their area and population size (mean and range) and refer to whether they are homogeneous or heterogeneous in terms of SES composition.
L152: The authors refer to a previous community SES classification by Mullings et al (2013). Please explain why they did not use this previous classification and how it compares the classification obtained in this study.
L181: The authors refer they tested for the presence of interactions. Please clarify how this was tested and which interactions are they referring to: multiplicative or additive? Also, since the studies are quite different, is there any interaction between the three studies? Is the effect of deprivation on blood pressure the same across these three population samples?
L198: ‘…estimate the effects…’ – The avoid using terms like ‘effect’. This implies causality, which cannot be ascertained in a cross-sectional study. Check the entire manuscript.
L203: Please clarify the concept of ‘crossed-effects models’. I believe this is not familiar for most of the readers.
L208: The authors performed multiple imputation to deal with missing data. There are some points that should be clarified. First, please explain how multiple imputation reduces potential bias (mentioned in 216); since missingness does not seem to be at-random (‘significant difference in the proportion of persons within tertiles of the second PCA SES’), this idea should be further expanded. Second, it is important to provide a table (as supplementary table) with the list of variables and corresponding percentage of missing values that were used for the imputation procedures. Finally, I highly recommend the authors to run sensitivity analyses based on the non-inputed dataset, to show that their results and conclusion are not driven by the imputation procedure. This can be in supplementary material as well.
L231-235: I do not understand the added value of this analysis. Please remove or clarify.

RESULTS
I do not understand why the authors keep showing results for the community-level variables that were merged in the SES-indexes. Since their focus is on the community SES index (at least in the abstract authors only present results for the associations with the index), I highly recommend using those two variables, instead of its components. Or, if their target is not the SES-index, then, these individual variables should be thoroughly described and the rational for their inclusion as isolated factors should be carefully described.
L289: Also, the authors describe associations between blood pressure and individual-level variables like age, BMI etc. Since this study is focused on community level SES and its influence on blood pressure, this type of description should be removed.
L308: (β +2.6 [1.0, 4.2], p<0.01 for middle vs. lower tertile) – Please remove the plus sign for the beta coefficient,

TABLES
In my point of view this manuscript has an excessive amount of tables, probably because they present the associations and comparisons for all the variables of their dataset instead of focusing on the main community-level SES determinants. If possible, I recommend the authors to reduce the number of tables.

FIGURES
I do not see the point of figure 5 and 6 and corresponding analysis. Also, from these figures (whose y-axis is not comparable), it looks like the association between SBP and SES is stronger in women.

DISCUSSION
L367: Please provide a plausible explanation on why the effect of community SES is stronger among men. There are a number of papers about gender differences in neighbourhood effects.
I miss a discussion on why no association was observed for diastolic blood pressure.
L403: Please include the % participation of each study.

Experimental design

Presented above

Validity of the findings

Presented above

---

## Round 0.2 · Minor Revisions

Thank you for your detailed and constructive responses and revisions. The manuscript is a delight to read and I’m sure would promote much discussion, and I hope further study, of these associations. Thank you also for your patience during this challenging time. I hope that you and your families are safe.

There are still a few comments from one of the original reviewers that will each need to be addressed, along with some comments from me below. Many of my comments are editing suggestions and some of these are stylistic. You should feel entirely comfortable with rejecting any stylistic suggestions if you prefer the current version of your text.

As the reviewer notes, the manuscript is much improved and I really appreciate the considerable effort that has been invested in this research. I am optimistic that these final points can be addressed without too much effort and that the revised manuscript will be able to be accepted.

Before I go through the minor/specific comments, there are two slightly larger ones and a brief comment that doesn't require any changes.

First, thank you very much for providing the code in Stata, which answered many of my questions and which I think a small group of readers will be extremely appreciative of. For the sample size calculation, the original calculation was part of the design of the study and so shouldn’t normally be revised, but I appreciate that you have reconsidered the primary outcome here. To save readers from needing to check over the Stata code you provide for these points, can you please briefly describe the approach used to estimate the ICCs and the mean number of participants per cluster used around Lines 247–251, along with their values. As also noted by the reviewer, the “correlation coefficients” on Line 245 cannot be correlation coefficients, being outside of [-1,1]. It would be worth checking the modified text to make sure that, based only on information presented there, the reader would be able to obtain the sample sizes you present. The supplementary file S2 contains “power onecorrelation” commands (and not “power oneproportion” commands as stated on Line 246), so I am assuming the description on Lines 243–244 is correct and the correlation coefficients on Line 245 seem to be approximately out by an order of magnitude.

The issue of whether it is possible that the same people could have been participants in two (or three) of the studies for the main analyses wasn’t directly addressed as far as I could tell. If this was the case, the independence assumption would be violated from unmodelled repeated measures. If it would not be possible to identify such participants, this could be addressed as a limitation in the discussion section (around Lines 590–604), where you would note whether this would be likely or not and/or how many such participants could plausibly be included twice. I’m not concerned that this would change your results and with some reassurance that the risk of this involving any or more than a handful of participants is very low, this won’t have any real impact on the reader’s confidence in your findings from the main analysis.

I don’t think you have to make any changes to the missing data text, but I’m assuming you’ve considered the possibility of informative missing data, i.e. missingness based on BP status, and determined that this is unlikely in this age group when deciding that MAR (and MCAR) are the likely mechanisms (Line 308). You refer to this assumption later on Lines 594–595.

A few specific comments are below (suggested edits or additions are in upper case):

Line 27: This is stylistic, but I’d suggest “two” here rather than “2” (c.f. “three” earlier in the sentence). See also Line 137 for “2” and “1” following “three”.

Line 69: “…levels OF urbanization…”

Line 95: Missing period at the end of the sentence.

Line 157: There seems to be a spurious space in “74 %” (c.f. no space on Line 158 in “55%” and elsewhere).

Lines 181–182: Perhaps “…We provided MGI with EACH participant’s address or electoral district…”

Line 183: Is the double “of Jamaica” intended here?

Line 210: Is “or” in “number or reported household assets” supposed to be “of”?

Line 213: Comma after “For each study”.

Line 214: “Classification RULES for the locally developed questionnaires have been…” or “ClassificationS for the locally developed questionnaires have been…”?

Line 215: “…while classification RULES used…” or “…while THE classificationS used…”?

Line 230: Comma after “For these analyses”? C.f. “In Jamaica, community” on the following line and similar punctuation elsewhere.

Lines 235–236: Comma after “For the communities included in this study”?

Line 240: Comma after “Given that the analyses were conducted using previously collected data”?

Line 303: Technically Table S4 shows percentages and not proportions.

Line 305: If you don’t mind another stylistic suggestion, I’d use “versus” rather than “vs.” here.

Lines 313–314: “…Graham anD colleagues…”

Lines 340–341: When you say “four observations per covariate coefficient for binary models”, do you mean at least four events and four non-events (the usual way sample sizes are discussed for binary dependent variables, e.g. from Peduzzi, et al. and subsequent studies into the stability of logistic regression models) or four observations (as usual for continuous dependent variables) or something else?

Lines 349–250: There were no instances of “estat gof” in the code and this command does not run as a post-estimation command after mixed effects logistic regression in Stata as far as I recall.

Line 359: No space needed in “eigen value”. I’d also say “…had AN adjusted eigenvalue” here and on Line 361.

Line 368: Delete one of “on” or “for” in “…communities with high scores on for PCA-SES1…”

Line 371: Just the singular “Table” in “Tables 1…”, unless you mean to refer to multiple tables here (including supplementary table S2?)

Line 374: Perhaps a comma after “between studies”?

Lines 383–384: “…significant differences FOR the distribution of persons…”? (or “in”)

Line 385: You say “Additional data with mean values…” but Table S6 also includes categorical data where counts and percentages are presented.

Line 389: Table 2 shows percentages but not proportions (apologies for the pedantry!) I haven’t worried about the distinction when you’re referring to tests (a test of proportions and a test of percentages are the same) or when you’re referring to interpretations (a higher proportion and a higher percentage are the same), but when you’re pointing out statistics, I suggest using percentages when these are presented as values between 0 and 100.

Line 394: “…relationship BETWEEN systolic BP…”

Line 397: “…We also assessED” (c.f. Line 394).

Line 405: “…in addition to hour sex-specific…” should be “our”

Line 406: Hyphenate “study specific” as you do “sex-specific” on the previous line?

Line 408: “…showed A statistically significant inverse association…”

Line 412: “Among males, A significant…”

Line 415: Missing comma before “spline 2” in “…spline 1 (z-score < -1) spline 2…”

Lines 419–420: “…for THE middle third among males and THE upper third among females…” (c.f. wording on Line 428)

Line 421: “…the 3-4 times/week CATEGORY compared to the [note doubled words here] to the ≤2 times/week CATEGORY for females only.”

Line 429: “…upper third of PCA-SES1 scoreS”

Line 435: If you’re concerned about the interpretation of “slightly reduced” here, you could say “slightly attenuated” to indicate it moved towards zero (as you use on Line 440). “attenuating” could also be used in place of “falling” on the following line when the coefficient technically increases towards zero.

Line 437: Perhaps “…resulted in the MAGNITUDE OF THE coefficient increasing…”?

Line 438: Perhaps “to” rather than “at” here in “…at -1.49…”

Line 441: Perhaps “reduced in magnitude” rather than “fell” here (from -1.30 to -0.46).

Line 442: “THE addition of BMI…”

Line 449: Perhaps a comma before “higher PCA-SES2 score” here?

Line 453: “Among males, A significant inverse association…”

Line 455: Note that you included an equals sign after beta just above on Line 450. See also Line 456.

Line 455: Perhaps a comma after “Among females”?

Line 455: “Among females, THERE WAS a…”

Line 460: “…THE odds ratio…”

Line 461: “upper third of PCA-SES1 was 0.67 (p=0.051), suggesting that higher SES was associated with” but given the level of significance implied on Line 246, which is a detail that should be repeated in the statistical methods section for clarity, this result isn’t statistically significant. The wording here needs some attention to avoid claiming too much from this result. While you add “borderline significant” on Line 462, this phrasing is definitely out of fashion these days as well as being slightly ambiguous about which side of the border the result would fall on. Personally, I’m fine with “non-statistically significant tendency” as long as this comes before the p-value and providing the fact that such associations are described in this way is mentioned in the statistical methods (with some appropriate caveat included, this result is certainly worth noting for future study but doesn’t achieve the requisite level of evidence in the present study).

Line 462: “TheRE were…”

Line 498: This is up to you, but there has been a movement away from terms like “hypertensives” towards “people with hypertension”, etc.

Line 506: Perhaps a comma after “In that study”?

Line 507: Another possible proportion/percentage clash for me (assuming this difference is absolute and not relative).

Lines 510–511: I suggest “lower” rather than “reduction in” in “…with 5.7 mmHg reduction in SBP…” as the latter can be misinterpreted as referring to changes rather than differences.

Line 518: “…reported A significant…”

Line 524: Perhaps a comma after “In this study”?

Lines 525 and 526: I don’t think you need the “previously” and “previous” here. Perhaps one would suffice?

Line 531: Perhaps a comma after “diastolic BP”?

Line 555: “…was associated WITH increased risk…”

Line 556: The multiplicities of “an” and “communities” don’t match, perhaps delete “an”.

Line 557: “…living IN poorer…”

Line 557: Hyphenate “tension prone”?

Line 559: “…from lower SES FOR males…” (or “in”, c.f. Line 560)

Line 565 “…group level effects beyond THOSE due to risk factors…”

Line 567: “…and ARE ultimately expressed…”

Line 568: Perhaps a comma after “With regards to blood pressure”?

Lines 581-582: Rather than “sufficiently large sample size”, which is vague and context-dependent, I suggest emphasising the widths of confidence intervals. If these are sufficiently precise, this would be a strength of your study, but this will be a judgment you need to make.

Line 616: Perhaps “…that interventionS to address…”?

Table 1: Capitalise and hyphenate “Chi-squared tests”? Same in Tables 2 and S6.

Table S2: Note that IQRs are single values (being the difference between the 75th and 25 percentiles, https://en.wikipedia.org/wiki/Interquartile_range) and you have presented the 25th and 75th percentiles here (and labelled this correctly in the table note, but with a reference to IQR there too). This is fine, many people find these more useful, but the statistics presented should match their description one way or another. Could you also add the statistical test used to obtain the p-values as a note to the table (you do this for other tables with more complex models)?

Tables S5, S8, S9, S10, S13, S14, and S15, and perhaps S7, S11, and S12, although you could argue that the splines should be treated differently, I would have recommended Wald tests for categorical variables with more than two levels so that the p-values are at the variable level and don’t depend on the choice of reference level, but the current approach is also fine.

Reviewer 3 ·

Basic reporting

shown in General comments for the author

Experimental design

shown in General comments for the author

Validity of the findings

shown in General comments for the author

Additional comments

The manuscript has significantly improved since its last version and my concerns have been addressed. However, before publication a couple of issues (listed below) need to be considered:

PAGE 13: ‘Correlation coefficients were -1.5 for females and -1.4 for males’ – Correlation coefficients vary from -1 to 1. Thus, this sentence has to be revised/clarified.

PAGE 19: ‘…therefore used linear splines in the bivariate and multivariable analyses…’
This new version of the manuscript uses splines to categorize the 2nd component of the SES index (I am personally not familiar with this statistical technique). This method has to be fully described and justified in the methods section, otherwise, it is very difficult to understand the results section.

TABLES:
In the second row from Tables 3 and 4, the authors refer to PCA SES Component 1 (z-score) but I believe they are referring to Component 2.

In my opinion, this manuscript still has an excessive amount of tables.

---

## Round 0.3 · accepted · Accept

Thank you for your constructive revisions and responses. I am delighted to accept your manuscript and look forward to seeing it in its final form in the not too distant future.